# NAC Transcription Factor *GmNAC12* Improved Drought Stress Tolerance in Soybean

**DOI:** 10.3390/ijms231912029

**Published:** 2022-10-10

**Authors:** Chengfeng Yang, Yanzhong Huang, Peiyun Lv, Augustine Antwi-Boasiako, Naheeda Begum, Tuanjie Zhao, Jinming Zhao

**Affiliations:** 1National Center for Soybean Improvement, Key Laboratory of Biology and Genetics and Breeding for Soybean, Ministry of Agriculture, State Key Laboratory of Crop Genetics and Germplasm Enhancement, Nanjing Agricultural University, Nanjing 210095, China; 2National Forage Breeding Innovation Base (JAAS), Key Laboratory for Saline-Alkali Soil Improvement and Utilization (Coastal Saline-Alkali Lands), Ministry of Agriculture and Rural Affairs, Institute of Animal Science, Jiangsu Academy of Agricultural Sciences, Nanjing 210014, China; 3Crops Research Institute, Council for Scientific and Industrial Research, Kumasi AK420, Ghana

**Keywords:** *Glycine max*, NAC transcription factors, drought tolerance, CRISPR/Cas9, protein interaction

## Abstract

NAC transcription factors (TFs) could regulate drought stresses in plants; however, the function of NAC TFs in soybeans remains unclear. To unravel NAC TF function, we established that *GmNAC12*, a NAC TF from soybean (*Glycine max*), was involved in the manipulation of stress tolerance. The expression of *GmNAC12* was significantly upregulated more than 10-fold under drought stress and more than threefold under abscisic acid (ABA) and ethylene (ETH) treatment. In order to determine the function of *GmNAC12* under drought stress conditions, we generated *GmNAC12* overexpression and knockout lines. The present findings showed that under drought stress, the survival rate of *GmNAC12* overexpression lines increased by more than 57% compared with wild-type plants, while the survival rate of *GmNAC12* knockout lines decreased by at least 46%. Furthermore, a subcellular localisation analysis showed that the *GmNAC12* protein is concentrated in the nucleus of the tobacco cell. In addition, we used a yeast two-hybrid assay to identify 185 proteins that interact with *GmNAC12*. Gene ontology (GO) and KEGG analysis showed that *GmNAC12* interaction proteins are related to chitin, chlorophyll, ubiquitin–protein transferase, and peroxidase activity. Hence, we have inferred that *GmNAC12*, as a key gene, could positively regulate soybean tolerance to drought stress.

## 1. Introduction

As the world’s population continues to rise, there is a growing requirement for increased crop yield in order to ensure agricultural production. Consequently, crop growers have been under much stress due to biotic and abiotic pressures [1]. Abiotic stresses, described as unfavourable non-living environmental factors, such as drought, salinity, flooding, and extreme temperatures, have a negative impact on crop plant survival and productivity [2]. Drought is considered to become the most serious hazard of the abiotic stressors, posing difficult obstacles to yield performance and productivity around the world [3].

Soybean (*Glycine max* [L.] Merr.) is one of the world’s most economically significant crops. Soybeans are used for human food and livestock feed and are highly valued for their high protein and oil contents and their numerous uses in industrial products [4,5,6]; however, soybean production and quality are threatened by multiple abiotic stressors including drought, salinity, and extreme temperatures. Drought is becoming a major constraint for crop production throughout the world. In soybean, drought stress mainly occurs during the growing season, leading to considerable yield loss and quality deterioration, mainly in arid and semi-arid zones. To adapt to the fast-changing climatic conditions and the prevailing threats caused by drought, many plants have established a variety of defence techniques ranging from altering the architecture of the roots and leaves to changing the metabolite composition and controlling the expression of genes involved in resistance to drought [7,8]. Recently, Huang et al. [9] reported that advancements in genomics and molecular biological approaches are useful for discovering stress-related genes, improving the soybean’s ability to adapt to environmental changes, and providing critical genetic resources in soybeans.

Several recent studies have shown that a complex network consisting of diverse proteins and metabolites is deployed in plant defence [10]. Transcription factors (TFs) are known for their crucial roles among the established participants in promoting plant responses to environmental stresses [11]. Some of these TF family (NAC, AP2/EREBP, WRKY, bZIP, bHLH, etc.) members are involved in response to adaptive stress, regulating stress-related genes by influencing the plant capacity in responding to stressful conditions, while other members engage in coordinating the growth and development of plants [12,13,14]. NAC proteins are composed of an extremely preserved NAC domain that consists of approximately 150 amino acids at the N-terminus and a highly variable transcription region on the C-terminus [15,16]. The NAC domain is classified into five subdomains ranging from A to E. Although C and D are highly conserved, positively charged subdomains associated with DNA binding [17,18], the function of subdomain A may be involved in the formation of functional dimers and subdomains B and E may play a role in the diversified function of the protein [19,20,21].

Several plant species, especially the terrestrials, are known to have NAC TF families that are extensively dispersed [22]. Since the initial discovery of NAC TFs in Arabidopsis thaliana [23], findings of NAC TFs have progressed rapidly, and their presence has been detected in rice soybeans, as well as in other plants [24,25]. Moreover, previous studies have demonstrated that the NAC TF family participates in responses to stress caused by biotic and abiotic factors, hormone signal transduction pathway, and cell apoptosis [26,27,28], and it plays important regulatory roles in various processes, including plant cell secondary wall formation, plant senescence, and lateral root development [29,30]. For instance, some researchers enhanced plant tolerance of drought stress by overexpressing Arabidopsis NAC TF family genes [31,32]. In rice (*Oryza sativa*), overexpression of *OsNAC5*, *OsNAC6*, *OsNAC10*, and *SNAC1* genes improved drought tolerance [33,34,35,36]. Furthermore, the overexpression of *GmSNAC49* in Arabidopsis improved drought tolerance by upregulating the genes related to drought and the ABA signal pathway [37]. In addition, *MusaNAC042* positively regulates drought and salinity tolerance in bananas [38], and *MusaSNAC1* improves drought tolerance by modifying stomatal closure and H_2_O_2_ content [27]. Recently, Ju et al. [26] found that overexpression of *VvNAC17* in transgenic Arabidopsis increases resistance to drought, salinity, and freezing, and upregulates the expression of ABA- and stress-related genes. 

Recently, there has been growing interest in the characterisation of the NAC TFs in soybean, emphasising stress responses. The overexpression of *GmNAC8* in soybean plants improves tolerance to drought stress by interacting with a drought-induced protein (GmDi19-3) [39]. In addition, the overexpression of *GmNAC109* in Arabidopsis improves tolerance to drought and salt stress by upregulating the expression of stress-related genes [40]. Another study reported that soybean-dehydration-induced *GmNAC085* plays a positive role in regulating plant drought tolerance [24]. In addition, the transient expression of *GmNAC065* and *GmNAC085* induce the appearance of leaf senescence features, which include loss of chlorophyll, yellowing of leaves, peroxidation of liquids, and the accumulation of H_2_O_2_ [41]. This evidence suggests that NAC TFs play significant and crucial roles in improving the growth of soybean plants and increasing their tolerance to biotic and abiotic stress. Therefore, to explore excellent germplasm resources, it is necessary to understand the function of NAC and its mechanism in the soybean. 

Clustered regularly interspaced short palindromic repeats/CRISPR-associated protein 9 (CRISPR/Cas9) system-mediated genome editing technology, which has been applied to various plants, has developed rapidly over the last few years and was successfully carried out in the soybean for the first time in 2015 [42]. Thus far, the CRISPR/Cas9 system has been used in soybeans for genetic transformation and hairy root genetic transformation to validate the functions of various genes related to diverse traits of interest in soybeans [43]. Successful application of the CRISPR/Cas9 system results in induced targeted mutagenesis of *GmFT2a*. Although *GmFT2a* regulates flowering in soybeans, the homozygous *GmFT2a* mutant delayed flowering [44]. These studies confirm the efficacy of applying the CRISPR/Cas9 system. The system is a highly specific and simple genome editing tool conducive to the functional verification of soybean genes and provides excellent potential for creating new soybean germplasm resources.

Previously, *GmNAC12* (*Glyma.16G043200*) in soybean plants has been reported with induced upregulated expression from stress caused by drought [25,45]. Our results showed that *GmNAC12* could positively regulate the tolerance of soybean to drought stress. The functional analysis of the *GmNAC12* gene could provide a basis for the response mechanism of soybean under drought stress, as well as provide a theoretical basis and germplasm resources for crop resistance genetic engineering breeding. In addition, using the yeast two-hybrid (Y2H) assay, we found that *GmNAC12* interacts with many functional proteins related to chitin, ubiquitin-protein transferase, peroxidase activity, etc. These results provided evidence support for the study of the function of soybean *GmNAC12* protein and the regulatory pathways involved and were conducive to the analysis of the regulatory network of *GmNAC12* in soybean stress tolerance. Therefore, our findings indicate that *GmNAC12* is a key regulator in the soybean plant.

## 2. Results

### 2.1. GmNAC12 Responded to Drought Stress as Well as ABA and ETH Hormone Treatments

To investigate the expression of *GmNAC12* in the leaves, roots, land cotyledons, and stems of soybean plants, we implemented the qRT-PCR assay. The results showed that *GmNAC12* was expressed in all the tissues, but the expression level of *GmNAC12* was highest and lowest in the cotyledons and the stems, respectively (Figure 1A). To explore the consequences of drought stress on the expression of *GmNAC12*, the soybean seedlings were exposed to drought conditions, and the gene’s expression was estimated by qRT-PCR. The findings indicated that the expression of *GmNAC12* in roots, stems, and leaves changed significantly and was significantly upregulated after 3, 5, and 7 days of drought stress, respectively (Figure 1B). These results indicate that *GmNAC12* may be involved in regulating drought stress.

Further, to investigate whether *GmNAC12* is involved in phytohormone regulation signals, qRT-PCR was used to detect the transcription level of *GmNAC12* in soybean seedlings after treatments with ABA and ETH. The results revealed that after the ABA treatment, the expression level of *GmNAC12* was significantly upregulated at 3 h and 12 h (Figure 1C), while under ETH treatment, the expression of *GmNAC12* was significantly upregulated at 3 h (Figure 1D). The results indicate that *GmNAC12* is also involved in the ABA and ETH regulation pathways.

### 2.2. Targeted Mutagenesis of GmNAC12 Produced Using the CRISPR/Cas9 System

The analysis of the gene structure demonstrated that the *GmNAC12* gene is made up of three exons and two introns, totalling 2348 bp. At the second exon, a site was designed to target mutagenesis (Figure 2A), and its matching CRISPR/Cas9 vector was converted into the soybean cultivar Tianlong 1 soybean cultivar for *GmNAC12* gene knockout using *Agrobacterium*-mediated soybean genetic transformation. The transformation resulted in 42 T_0_ generations of *GmNAC12* transgenic lines with a transformation efficiency of 5.6%. Among them, 26 lines were *GmNAC12* gene knockout lines with an edited rate of 61.9%.

After analysing 26 *GmNAC12* gene knockout lines, five gene editing types were identified (Figure 2B): *GmNAC12-KO1* (deletion 1 bp), *GmNAC12-KO2* (deletion 2 bp), *GmNAC12-KO3* (3 bp deletion), *GmNAC12-KO4* (4 bp deletion), and *GmNAC12-KO5* (5 bp deletion). In comparing these with the amino acid sequence of wild-type soybeans, it was noted that frameshift mutations occurred in *GmNAC12-KO1*, *GmNAC12-KO2*, *GmNAC12-KO4*, and *GmNAC12-KO5*, which lead to early termination of translation. *GmNAC12-KO3* lacks 3 bp in the amino acid sequence, resulting in one amino acid change and one amino acid deletion (Figure 2C).

### 2.3. GmNAC12 Enhanced Drought Tolerance in Soybean Plants

We generated *GmNAC12* overexpression (OE1 and OE4) and *GmNAC12* knockout (KO1 and KO2) lines to investigate the reaction of *GmNAC12* when exposed to drought conditions. *GmNAC12* overexpression lines (OE1 and OE4), gene knockout lines (KO1 and KO2), and wild-type plants were exposed to drought treatment (stress) after 14 days of standard growth. After 10 days of drought conditions, the leaves of the *GmNAC12* gene knockout lines (KO1 and KO2) were severely wilted compared to the leaves of the wild-type plants; however, the *GmNAC12* overexpression lines (OE1 and OE4) only showed minor wilting symptoms (Figure 3A). In addition, after rewatering, only 59% of the wild-type plants recovered, while 92% and 94% of the OE1 and OE4 *GmNAC12* overexpression lines recovered, respectively, and they exhibited a significantly higher survival rate compared to the wild type. Furthermore, 29% and 33% of the *GmNAC12* gene knockout lines KO1 and KO2 recovered, respectively, and the survival rate was significantly lower than the wild type (Figure 3B). The above results indicate that *GmNAC12* plays a positive role in regulating soybean drought tolerance.

### 2.4. GmNAC12 Encoded a Nuclear Localisation Protein

In establishing the subcellular localisation, the domain and nuclear localisation signal of the *GmNAC12* protein were predicted online. An amino acid sequence analysis showed that *GmNAC12* encoded 353 amino acids, with a NAM domain at 9–138 amino acids (Appendix A) and two NLSs, RPKRQVSNMDEETLYPSKKYLSS and RPKRQVSNMDEETLYPSKKYLSSS (Appendix A), at 291–314 amino acids. It was predicted that *GmNAC12* is a nuclear localisation protein.

To verify the prediction results of the subcellular localisation vector of *GmNAC12*, namely, *pBinGPF4-GmNAC12*, it was constructed, as shown in Appendix A. The Agrobacterium-mediated transformation of tobacco transferred the expression vector into tobacco leaves for transient expression. The green fluorescent protein signal was observed through laser confocal microscopy. The results showed that a green fluorescent protein signal of an empty vector was detected on the cytoplasm, cell membrane, and nucleus (Figure 4E–H). In contrast, the fusion protein expressed by *pBinGPF4-GmNAC12* was only detected in the nucleus (Figure 4A–D). These results confirm that the *GmNAC12* protein is a nuclear-localised protein.

### 2.5. Candidate Interaction Protein Analysis of GmNAC12

The study of interaction proteins analyses the action mechanism of the targeted protein and helps explore new functions of the protein [46]. To further understand the action mechanism of *GmNAC12*, we used yeast two-hybrid technology to screen the interaction proteins of *GmNAC12*. After screening and identification, 185 candidate proteins were determined to interact with *GmNAC12*.

The 185 candidate interaction proteins of *GmNAC12* were annotated with GO functions (Figure 5A, Appendix A). The identified interaction proteins related to one or more of the following three GO categories: biological processes (BP), cellular components (CC), and molecular function (MF). An analysis of the biological processes indicated that the interaction proteins were mainly involved in translation (GO: 0006412), fatty acid beta-oxidation (GO: 0006635), plant-type secondary cell wall biogenesis (GO: 0009834), hydrogen peroxide catabolic process (GO: 0042744), defensive responses to the bacterium (GO: 0042742), defensive responses to the virus (GO: 0051607), and defensive responses to fungus (GO: 0050832) and other processes. An analysis of cell composition indicated that the interaction proteins were mainly composed of the cytosolic small ribosomal subunit (GO: 0022627), ribosome (GO: 0005840), photosystem I (GO: 0009522), and photosystem II (GO: 0009523). Furthermore, a molecular biological function analysis determined that the interaction proteins mainly included structural components of the ribosome (GO: 0003735), chitin-binding activity (GO: 0008061), chlorophyll-binding activity (GO: 0016168), ubiquitin-protein transferase activity (GO: 0004842), and peroxidase activity (GO: 0004601), among others.

On the basis of the results of previous studies, we know that secondary metabolites and signals mediated by hormones perform crucial roles in improving stress tolerance. Considering this knowledge, we performed a KEGG enrichment analysis to broaden our understanding of the metabolic pathways of genes in plant cells. The KEGG enrichment analysis of the 185 candidate interaction proteins of *GmNAC12* indicated that the proteins mainly involved 15 pathways (Figure 5B, Appendix A), including ribosomes (ko03010), photosynthesis-antenna proteins (ko00196), drugs metabolic-other enzymes (ko00983), photosynthesis (ko00195), endocytosis (ko04144), phenylpropanoid biosynthesis (ko00940), and other regulatory pathways.

### 2.6. Expression Analysis of GmNAC12 under Biotic-Stress-Related Hormones

On the basis of the GO and KEGG analyses, we speculated that *GmNAC12* might also be involved in biotic stresses. Previous studies have shown that SA and MeJA are the key phytohormone signals for plants to respond to biotic stress. The expression levels of *GmNAC12* in diverse tissues of soybean seedlings under SA and MeJA phytohormone treatments were detected by qRT-PCR. Under the SA treatment, the expression of *GmNAC12* was significantly upregulated at 3 h (Figure 6A). Under the MeJA treatment, the expression of *GmNAC12* was significantly downregulated at 1 h, then significantly upregulated at 12 h (Figure 6B). In summary, *GmNAC12* might help regulate soybean response to biotic stress in a way that depends on plant hormones.

## 3. Discussion 

In the context of rapidly changing climate, drought is considered one of the major environmental factors limiting plant growth and development, which adversely affects a plant’s yield. However, many strategies have been devised and implemented to prevent drought’s damaging effects on plant growth, thus improving productivity. As upstream genes for gene expression regulation, the NAC TF family can respond quickly when plants are subjected to unfavourable environmental pressures [20]. Hence, the functional identification of NAC TFs in soybean plants is important for studying how the soybean adapts to drought stress. From this perspective, we analyzed the expression of *GmNAC12* in the tissues (roots, stems, and leaves) of soybean seedlings after drought stress by qRT-PCR and noted substantial upregulation of *GmNAC12* in the various tissues (Figure 1B). The present results support the findings of previous researchers [25,45], which suggest that *GmNAC12* may regulate drought tolerance in soybeans.

The ABA and ETH regulation pathways are often involved in abiotic stress in plants [47,48]. Under drought stress conditions, the concentration of ABA in the plant increases, which promotes stomata closure and reduces transpiration, ultimately enhancing the drought resistance of plants [49]. In this study, *GmNAC12* was significantly upregulated under the induction of ABA and ETH (Figure 1C,D). The results are similar to the findings reported by Yang et al. [39] in that *GmNAC8*, a *GmNAC12* homologous gene, is induced by drought, ABA, and ETH. Therefore, we inferred that *GmNAC12* may regulate drought stress through plant hormones’ signal transduction pathway.

Since the advent of CRISPR/Cas9, genome editing has enabled quick and easy transformations in plants to characterise gene functions and improve traits, mainly through double-strand breaks generated by CRISPR/Cas9 to induce mutations [50]. Due to the advantages of using CRISPR/Cas9 technology to knockout, insert, or replace important genes in plants, precise improvements of plant traits or varieties can be achieved to cultivate new varieties suitable for various geographical and environmental conditions, which can help to alleviate the current soybean dilemma. The realisation of soybean gene editing depends on soybean genetic transformation technology; therefore, establishing a stable and efficient soybean genetic transformation system is essential. Applying gene editing technology to soybeans provides an essential tool for analysing soybean gene function and molecular mechanisms [51]. 

In this study, to explore the specific function of *GmNAC12* under drought stress, five different *GmNAC12* gene knockouts were created by the CRISPR/Cas9 system. Of the five gene knockouts, *GmNAC12-KO1*, *GmNAC12-KO2*, *GmNAC12-KO4*, and *GmNAC12-KO5* had frameshift mutations due to base deletion (Figure 2B), which led to early termination of translation and resulted in the almost complete deletion of the transcriptional regulatory domain at the C-terminal of *GmNAC12* (Figure 2C), as well as loss of function [20]. The remaining knockout gene, *GmNAC12-KO3*, lacked three bases, which resulted in an amino acid deletion and an amino acid mutation; however, its function in the transcriptional regulatory activity of *GmNAC12* remains unknown.

Through the Agrobacterium-mediated genetic transformation of the soybean, three stable *GmNAC12* overexpression lines were obtained after screening and identifying. Drought treatment was performed on the *GmNAC12* gene of overexpression, gene knockout, and wild-type plants (Figure 3A). The treatment findings demonstrated that the survival rate of the *GmNAC12* overexpressed lines was greater than that of the wild-type plants, and the plants showed more substantial drought tolerance. Conversely, the survival rate of the *GmNAC12* gene knockout plants was lower compared to the survival rate of the wild-type plants, indicating limited drought tolerance (Figure 3B). These results indicate that *GmNAC12* positively regulates drought tolerance in soybeans, which is consistent with recent findings indicating that GmNAC8, the homologous gene of *GmNAC12*, positively regulates the drought stress tolerance of soybeans [39].

We used *GmNAC12* as the bait protein and screened 185 candidate interacting proteins through yeast two hybrid (Y2H). Interestingly, although *GmNAC12* was a transcription factor expressed in the nucleus (Figure 4), its interacting proteins were expressed not only in the nucleus but also in the cytoplasm and cell membrane. It was shown that the rose transcription factor PTM could change its expression position in response to drought stress [52], which might suggest that *GmNAC12* was not exclusively expressed in the soybean nucleus. In addition, GmHsp90A2 could change its protein expression region by interacting with GmHsp90A1 in soybean [9]. Therefore, *GmNAC12* might change its expression position by interacting with proteins that function in different regions of the cell.

Furthermore, we performed a GO annotation of 185 interaction proteins of *GmNAC12* and found that some interaction proteins related to *GmNAC12* functions involve peroxidase activity (Figure 5A). The regulation of drought stress in plants involves a complex signal transduction network, and reactive oxygen species (ROS) have a signal interface function in plants adapting to drought stress [53]. Previous studies on *GmNAC8* showed that it could improve the soybean’s tolerance of drought by increasing the activity of intracellular peroxidase to remove excess ROS [39]. During this process, peroxidase can partially eliminate the damage plants under drought stress experience due to ROS accumulation [15,53], thereby improving the drought tolerance of *GmNAC12* in overexpressed plants. Therefore, the action mechanism of *GmNAC12* that enhances the drought stress tolerance of soybeans by removing excessive ROS may be the same as that of *GmNAC8*.

NAC transcription factors play important roles in the various signalling pathways that govern how a plant responds to biotic and abiotic stress and developmental activities [20,54]. Many studies have reported that SA and MeJA are the main signals used in the system to obtain resistance signal pathways [55]. The current work exhibited that *GmNAC12* expression was considerably upregulated by the induction of the plant hormones such as SA and MeJA (Figure 6A,B). As a result of binding protein interactions, during which *GmNAC12* interacts with proteins involved in responses to bacteria, fungi, and viruses, we speculate *GmNAC12* could participate in regulating biotic stresses by using a soybean’s plant hormone pathway. Additionally, *GmNAC12* interacts with proteins involved in plant secondary wall synthesis and photosynthetic systems, indicating that *GmNAC12* may also be a key regulator for improving plant growth and development [56]. We are certain that the *GmNAC12* gene in soybeans controls functions related to abiotic stresses such as drought and biotic stresses caused by pathogens such as bacteria, fungi, and viruses.

## 4. Materials and Methods

### 4.1. Plant Materials, Growth Conditions, and Stress Treatments

The soybean cultivar, Tianlong 1, was used as the plant material in various experiments. Soybean seeds were cultivated in the soil in a controlled growth chamber at a temperature of 25 °C under long day (16/8 h light/dark) conditions and a relative humidity of 70%. Additionally, various stress treatments were imposed on the plants when they reached the first trifoliate growth stage. We initiated the drought treatment of soybean seedlings by depriving the plants of water for 0, 1, 3, 5, and 7 days, and we harvested the roots, stems, and leaves at different time intervals. For various hormone treatments, soybean seedlings were transferred in 1/2 Hoagland nutrient solution with 100 mM MeJA (methyl jasmonate), 2 µM ETH (ethylene), 2 mM SA (salicylic acid), or 150 µM ABA (abscisic acid). The roots were harvested at 0, 1, 3, 6, 12, and 24 h time intervals and were immediately preserved in liquid nitrogen for further investigation.

### 4.2. Quantitative Real-Time PCR Assay

Total RNA was obtained from the samples using an RNAprep Pure Plant Kit (Tiangen, Beijing, China). The total RNA’s purity and concentration were established using a Nanodrop UV spectrophotometer (Thermo Fisher Scientific, Waltham, USA) and an RNA Nanochip on an Agilent Bioanalyzer 2100 (Agilent Technologies, Palo Alto, CA, USA). The Prime Script^TM^ RT Reagent Kit (TaKaRa, Kyoto, Japan) was used for cDNA synthesis. A quantitative real-time PCR (qRT-PCR) assay was performed for each cDNA template following the standard protocol using AceQ qPCR SYBR Green Master Mix (Vazyme, Nanjing, China). In addition, *GmActin11* (*Glyma.18g290800*) was used as an internal control by following the Ct-method to normalise expression levels [57]. NCBI Primer-BLAST designed all the primers used, and all primers are listed in Appendix A. Three biological replications were used for qRT-PCR assays, and three measurements were performed on each replicate.

### 4.3. Subcellular Localisation of GmNAC12 Protein

The nuclear localisation signal (NLS) of *GmNAC12* was predicted through the online software cNLS Mapper (http://nls-mapper.iab.keio.ac.jp/cgi-bin/NLS_Mapper_form.cgi) using its amino acid sequence, which accessed on 1 June 2019. To verify the localisation of the *GmNAC12* protein in plant cells, we constructed a fusion vector using the *pBIN-GFP4* vector. Following the manufacturer’s protocol, we cloned the CDS of the *GmNAC12* gene into the pBIN-GFP4 vector, minus the terminator (Appendix A), then injected the constructed and empty vectors into tobacco leaves. The fluorescence of GFP was detected at 488 nm and 405 nm by an upright confocal microscope (Carl Zeiss, Thornwood, NY, USA) at 48–72 h post-inoculation.

### 4.4. GmNAC12 Knockout in Soybean Plants Using the CRISPR/Cas9 System

We used the CRISPR/Cas9 system to knockout *GmNAC12*. The CRISPR-P web tool targeted one site and was selected in the second exon of *GmNAC12* (Figure 3A). For the targeted knockout of *GmNAC12*, a 20 bp exon sequence from the target site of the *GmNAC12* gene was duplicated into CRISPR/Cas9 vector *p0645*. The soybean cultivar Tianlong 1 was used to transform the CRISPR/Cas9 knockout vector using the protocol established by Yang et al. [39]. Using the modified CTAB method, we removed the genomic DNA from the young leaves of soybean plants from the T_0_ generation. Subsequently, PCR amplified an area extending across the target site, and the PCR products were sequenced (GenScript, Nanjing, China). Appendix A lists the primers used in the study. For heterozygous mutations, overlapping peaks were found at the target site, while the wild-type and homozygous mutations did not record overlapping peaks at the target site. The homozygous mutations at the target site were discovered by using DNAMAN software to execute a sequence alignment with the wild-type sequence. For the detection of targeted mutations, the same protocol was used at T_1_ and T_2_ generations.

### 4.5. GmNAC12 Overexpression in Soybean Plants

The full-length CDS of the *GmNAC12* gene was cloned into the *pTF101.1* vector under the control of the *CaMV 35S* promoter. After the vector sequence was confirmed by sequencing, the recombinant *pTF101.1–GmNAC12* plasmid vector was transformed into soybean cultivar Tianlong 1 using *Agrobacterium tumefaciens* strain EHA101. The selection marker gene (*bar*) and a *35S* promoter were amplified by PCR to verify the positive transgenic plants. The homozygous transgenic lines’ phenotypic evaluation was conducted at the T_3_ generation. 

### 4.6. Yeast Two-Hybrid Assay

The whole coding sequence for the *GmNAC12* was cloned into vector *pGBKT7* to produce the recombinant *pGBKT7-GmNAC12* plasmid vector as bait while using the soybean yeast library plasmids as prey. Following the manufacturer’s protocol (Takara, Kyoto, Japan), the *pGBKT7-GmNAC12* construct and library plasmids were co-transformed into the Y2H Gold yeast strain. Subsequently, these yeast cells were streaked onto media containing SD/-Trp/-Leu plates, SD/-Trp/-Leu/-Ade/-His+AbA+10 mM 3-AT plates, and SD/-Trp/-Leu/-Ade/-His+AbA+10 mM 3-AT + X-α-Gal plates.

### 4.7. GO Annotation and KEGG Enrichment Analysis

The GO annotation comes from the GO (Gene Ontolog) database. The software Goatools (https://github.com/tanghaibao/GOatools), which accessed on 21 September 2021, was used for enrichment analysis, and the method was Fisher’s exact test. In controlling the calculated false-positive rate, the *p*-value was corrected using four multiple-test approaches: Bonferroni, Holm, Sidak, and the false discovery rate. Usually, when the *p*-value is lower than 0.05, the GO function is considered significant enrichment. The KEGG (Kyoto Encyclopaedia of Genes and Genomes) database assists with the systematic examination of gene roles, contact genomics, and functional information. Using the KEGG database, genes can be categorised by their pathways or by the functions they are involved in. The analysis of the KEGG uses KOBAS (http://kobas.cbi.pku.edu.cn/home.do), which accessed on 21 September 2021, to determine interacting proteins pathway enrichment analysis. To control the false positive rate calculation, the Benjamini–Hochberg (BH) method was used for many tests. The BH method sets the *p*-value at 0.05 as the threshold. When the KEGG pathway meets the set conditions, this is defined as significant enrichment.

The primers and sequences used in this study were from Genscript (Nanjing, China).

## 5. Conclusions

This study explored the potential functions of *GmNAC12*, a member of the soybean NAC family, in response to abiotic stress. The present results revealed that *GmNAC12* overexpressed lines demonstrated improved tolerance to drought compared to wild-type plants, whereas *GmNAC12* knockout lines were sensitive under drought conditions. Furthermore, the findings demonstrated that *GmNAC12* positively regulates drought stress in soybeans. In addition, the 185 candidate interaction proteins of *GmNAC12* identified by yeast two-hybrid assay were performed on GO analysis and KEGG enrichment, showing that some interaction proteins of *GmNAC12* are related to peroxidase activity. Hence, we inferred that *GmNAC12*, as a key gene, could positively regulate soybean tolerance to drought stress.

## Figures and Tables

**Figure 1 ijms-23-12029-f001:**
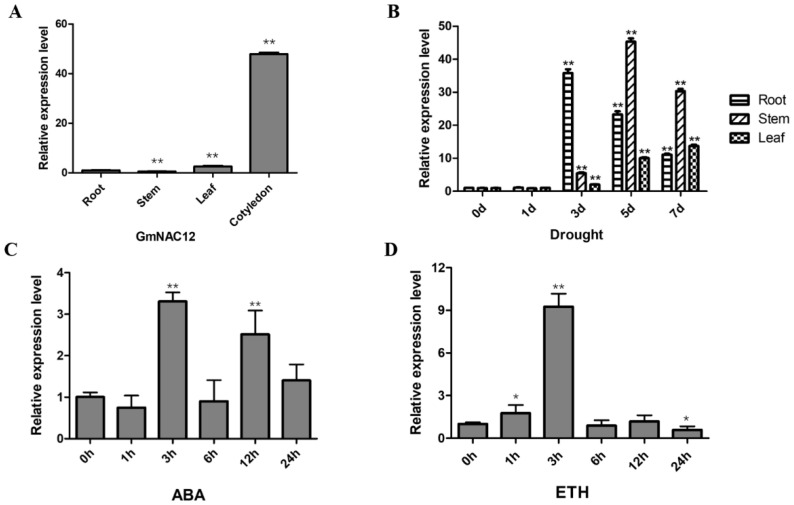
Expression pattern of *GmNAC12* under drought stress and ABA and ETH hormone treatment by qRT-PCR. (**A**) Expression of the *GmNAC12* in cotyledon, roots, stems, and leaves in normal condition by qRT-PCR. (**B**) Expression of the *GmNAC12* in roots, stems, and leaves under drought stress by qRT-PCR. (**C**,**D**) Expression of *GmNAC12* in roots under ABA (150 µM) and ETH (2 mM) treatments determined by qRT-PCR. The data represent the means ± SEs, *n* = 3. * *p* < 0.05 (Student’s *t*-test). ** *p* < 0.01 (Student’s *t*-test).

**Figure 2 ijms-23-12029-f002:**
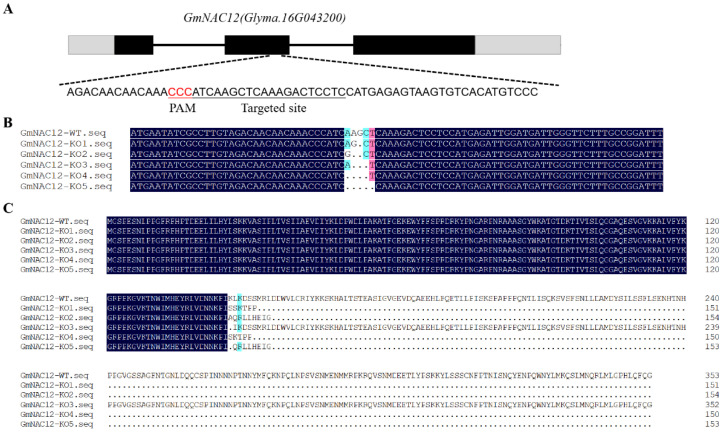
Targeted mutagenesis of *GmNAC12* induced by CRISPR/Cas9. (**A**) Gene structures of *GmNAC12* with a CRISPR/Cas9 target site designed in the second exon. Black stripe, black line, and grey stripe represent exon, intron, and UTR (untranslated regions), respectively. The underlined nucleotides indicate the target site. Nucleotides in red represent PAM sequences. PAM, protospacer adjacent motif. (**B**,**C**) DNA sequences and amino acid sequences of wild-type and representative mutation types, *GmNAC12-KO1*, *GmNAC12-KO2*, *GmNAC12-KO3*, *GmNAC12-KO4*, and *GmNAC12-KO5*, induced at the target site. Underline, insertions. Dashes, deletions.

**Figure 3 ijms-23-12029-f003:**
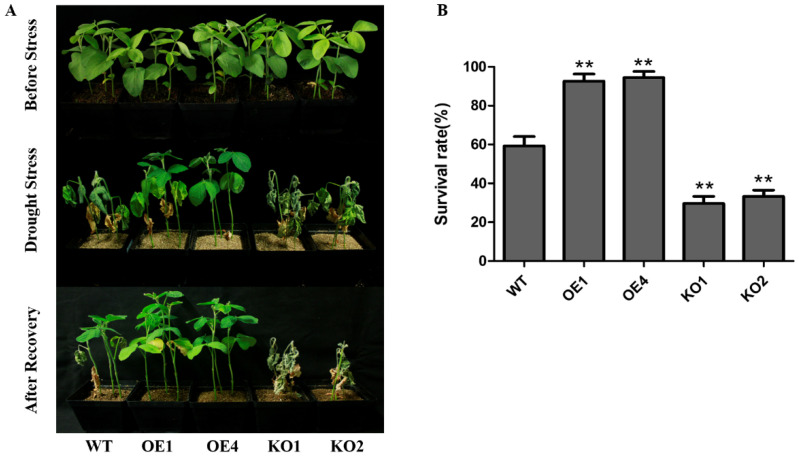
*GmNAC12* positively regulated drought tolerance in soybean. (**A**) Performance of wild-type (WT) plants, *GmNAC12* overexpression lines (OE1 and OE4), and *GmNAC12* knockout lines (KO1 and KO2) under drought stress and after recovery. (**B**) Survival rate of wild-type plants, *GmNAC12* overexpression lines, and *GmNAC12* knockout lines after recovery (*n* = 3). Over 30 plants in each line were used for survival rate analysis. The data represent the means ± SEs. ** *p* < 0.01 (Student’s *t*-test).

**Figure 4 ijms-23-12029-f004:**
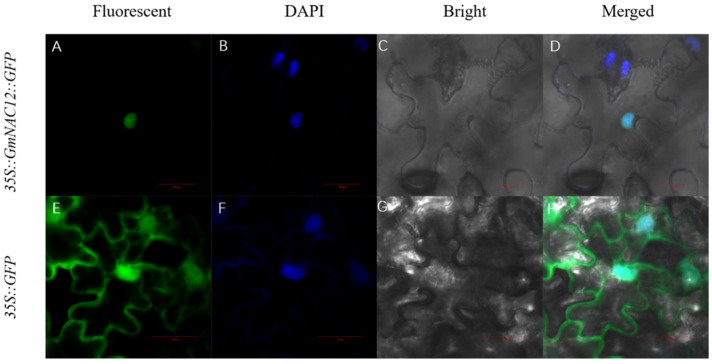
Subcellular localisation of *GmNAC12* in a tobacco cell. (**A**–**D**) *35S::GmNAC12::GFP* fluorescence images in a tobacco cell. (**E**–**H**) *35S::GFP* fluorescence images in a tobacco cell. *Nicotiana benthamiana* leaves were transiently infiltrated with A. tumefaciens EHA105-containing vector expressing *35S::GFP* or *35S::GmNAC12::GFP*. All images were collected using the Zeiss confocal microscope after agroinfiltration for 48 h. DAPI images indicate nuclear staining. Scale bars are 50 μm.

**Figure 5 ijms-23-12029-f005:**
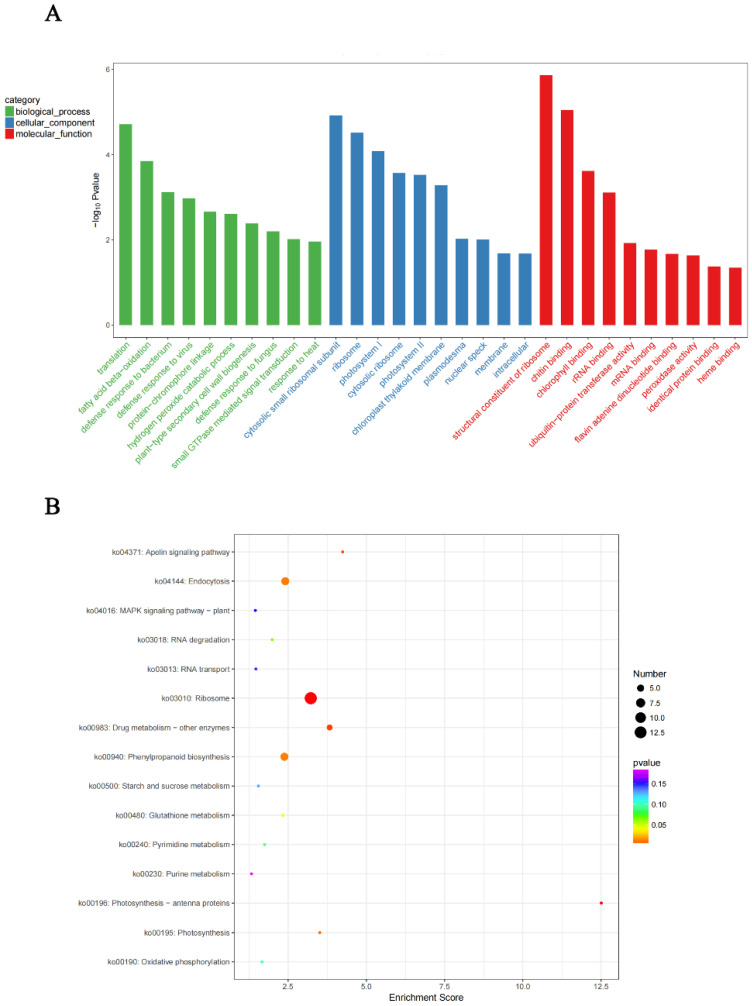
GO and KEGG analysis of candidate interaction proteins of *GmNAC12*. (**A**) Diagram showing GO enrichment analysis of candidate interaction proteins of *GmNAC12*. BP, biological processes; CC, cellular components; MF, molecular function. (**B**) KEGG analysis of candidate interaction proteins of *GmNAC12*.

**Figure 6 ijms-23-12029-f006:**
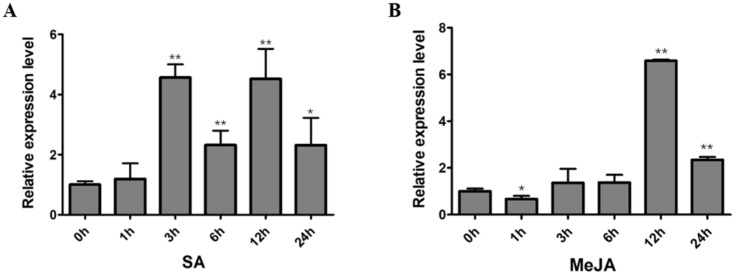
Expression of *GmNAC12* under SA and MeJA hormone treatments. (**A**,**B**) Expression of *GmNAC12* in roots under SA (2 mM) and MeJA (100 µM) treatments determined by qRT-PCR (*n* = 3). The data represent the means ± SEs. * *p* < 0.05 (Student’s *t*-test). ** *p* < 0.01 (Student’s *t*-test).

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
