# Peer review of "NAC Transcription Factor GmNAC12 Improved Drought Stress Tolerance in Soybean"

_ijms, 2022, doi:10.3390/ijms231912029_

Round 1

Reviewer 1 Report

Lines 2-3: Title

The title is too long and I suggest the following:

NAC Transcription Factor GmNAC12 Improved Drought Stress Tolerance in Soybean

Line 16: …regulate various stresses in plants; …

Talk about your stress, water stress (or drought) in abstract.

Line 17: In this research

After a brief introduction, you started talking about the research with results, while you should describe what you did at the first. We don’t know what your research is.

Lines 17-…:

Use past tense verbs to talk about your research activities.

we establish => established

is involved => was involved

is significantly up-regulated => was …

Line 18: from the soybean

Add the scientific name.

Lines 16-29: abstract

All the results presented in the abstract were descriptive, report some of the results quantitatively using numbers and percentages.

Line 30: keywords

Please select another words instead of GmNAC12; Drought stress

Line 44: excessive salinity => salinity

Soybean is not a tolerant plant to salt stress.

Lines 113-121: Herein, we selected … the soybean plant.

You should describe you aim at the end of introduction, but you summarize your results again. You should indicate the theory of this research and highlight the novelty.

Line 270: discussion

I suggest to remove the sub-headings in this section and To interpret the findings in a comprehensive discussion.

Line 359: … depriving the plants of water for 0, 1, 3, 5, and 7 days…

How did you select these levels? Are you sure that 7 days is a good level as a most severe water stress?

Line 359: … depriving the plants of water for 0 …

What does 0 mean? Daily irrigation?

Author Response

Dear reviewer

We appreciate the critical comments and useful suggestions from you. All the questions and comments have been acknowledged. The manuscript has been also rewritten and organized as suggestions. Herein we re-submit our manuscript entitled “A Stress-related NAC Transcription Factor GmNAC12 from Soybean Positively Regulates Drought Stress Tolerance” (Manuscript ID: ijms-1935753) for consideration as article by International Journal of Molecular Sciences.

Sincerely,

Prof. Jingming Zhao

Reviewer: Comments on the manuscript " A Stress-related NAC Transcription Factor GmNAC12 from Soybean Positively Regulates Drought Stress Tolerance ", by Yang et al.

  1. A Stress-related NAC Transcription Factor GmNAC12 from Soybean Positively Regulates Drought Stress Tolerance.

Re: Thank you. We have modified the title of “NAC Transcription Factor GmNAC12 Improved Drought Stress Tolerance in Soybean” according to your suggestion.

  1. Line 16: …regulate various stresses in plants; …

Talk about your stress, water stress (or drought) in abstract.

Re: Thank you. We have modified this sentence according to your suggestion.

“NAC transcription factors (TFs) could regulate various drought stresses in plants;”.

  1. Line 17: In this research

After a brief introduction, you started talking about the research with results, while you should describe what you did at the first. We don’t know what your research is.

Re: Thank you. We have added purpose of research in this sentence according to your suggestion.

“To unravel NAC TFs function, we established that GmNAC12, a NAC TF from the soybean (Glycine max;), was involved in the manipulation of stress tolerance”.

  1. Lines 17-…:

Use past tense verbs to talk about your research activities.

we establish => established

is involved => was involved

is significantly up-regulated => was …

Re: We have changed the relevant words.

To unravel NAC TFs function, we established that GmNAC12, a NAC TF from the soybean (Glycine max) was involved in the manipulation of stress tolerance. The expression of GmNAC12 was significantly up-regulated more than 10-fold under drought stress and more than 3-fold under abscisic acid (ABA) and ethylene (ETH) treatment.

  1. Line 18: from the soybean

Add the scientific name.

Re: Thank you. We have added scientific name there.

we established that GmNAC12, a NAC TF from the soybean (Glycine max;), was involved in the manipulation of stress tolerance.

  1. Lines 16-29: abstract

All the results presented in the abstract were descriptive, report some of the results quantitatively using numbers and percentages.

Re: Thank you. We have added relevant content to the abstract.

NAC transcription factors (TFs) could regulate drought stresses in plants; however, the function of NAC TFs in soybeans remains unclear. To unravel NAC TFs function, we established that GmNAC12, a NAC TF from the soybean (Glycine max) was involved in the manipulation of stress tolerance. The expression of GmNAC12 was significantly up-regulated more than 10-fold under drought stress and more than 3-fold under abscisic acid (ABA) and ethylene (ETH) treatment. In order to determine the function of GmNAC12 under drought stress conditions, we generated GmNAC12 overexpression and knock-out lines. The present findings showed that under drought stress, the survival rate of GmNAC12 overexpression lines increased by more than 57 % compared with wild-type plants, while the survival rate of GmNAC12 knockout lines decreased by at least 46 %. Furthermore, a subcellular localization analysis showed that the GmNAC12 protein is concentrated in the nucleus of the tobacco cell. In addition, we used a yeast two-hybrid assay to identify 185 proteins that interact with GmNAC12. Gene ontology (GO) and KEGG analysis showed that GmNAC12 interaction proteins are related to chitin, chlorophyll, ubiquitin-protein transferase, and peroxidase activity. Hence, we have inferred that GmNAC12 positively regulates drought stress probably by using the antioxidant enzyme defence pathway.

  1. Line 30: keywords

Please select another words instead of GmNAC12; Drought stress

Re: Thank you. We have replaced “GmNAC12” with “NAC transcription factors”, and “Drought stress” with “Drought tolerance”.

  1. Line 44: excessive salinity => salinity

Soybean is not a tolerant plant to salt stress.

Re: Thank you. We have changed the words.

“however, soybean production and quality are threatened by multiple abiotic stressors including drought, salinity, and extreme temperatures.”

  1. Lines 113-121: Herein, we selected … the soybean plant.

You should describe you aim at the end of introduction, but you summarize your results again. You should indicate the theory of this research and highlight the novelty.

Re: Thank you. We have changed end of introduction.

Previously, GmNAC12 (Glyma.16G043200) in soybean plants has been reported with induced up-regulated expression from stress caused by drought [25, 45]. Our results showed that GmNAC12 could positively regulate the tolerance of soybean to drought stress. The functional analysis of GmNAC12 gene could provide a basis for the response mechanism of soybean under drought stress, and provide a theoretical basis and germplasm resources for crop resistance genetic engineering breeding. In addition, using the yeast two-hybrid (Y2H) assay, we found that GmNAC12 interacts with many functional proteins related to chitin, ubiquitin-protein transferase, peroxidase activity, etc. These results provided evidence support for the study of the function of soybean GmNAC12 protein and the regulatory pathways involved, and were conducive to the analysis of the regulatory network of GmNAC12 in soybean stress tolerance. Therefore, our findings indicate that GmNAC12 is a key regulator in the soybean plant.”

  1. discussion

I suggest to remove the sub-headings in this section and To interpret the findings in a comprehensive discussion

Re: Thank you. We have removed the sub-headings in disscussion.

  1. Line 359: … depriving the plants of water for 0, 1, 3, 5, and 7 days…

How did you select these levels? Are you sure that 7 days is a good level as a most severe water stress?

Re: We chose the time point to start on the first day of drought treatment, and samples were taken every two days. By the seventh day, wild-type lines had already begun to show wilting phenotype. Wilted samples were present in one replicate, and the RNA extraction and quantitative PCR data of the treated samples were significantly different from those of the wilted treatment. So, the sampling stopped on the seventh day. Related soybean drought treatment methods have been published as “Yang C, Huang Y, Lv W, et al. GmNAC8 acts as a positive regulator in soybean drought stress[J]. Plant Science, 2020, 293:110442”.

  1. Line 359: … depriving the plants of water for 0 …

What does 0 mean? Daily irrigation?

Re: Drought day 0 means the last day of normal watering.

Reviewer 2 Report

In the paper entitled “A stress-related NAC transcription factor GmNAC12 from soybean positively regulates drought stress tolerance” Yang and co-workers produce and characterize a series of soybean lines either overexpressing GmNAC12 or harboring a knockout version of this gene to show the relevance of this transcription factor in drought stress tolerance. Furthermore, they explore the transcriptional response of GmNAC12 to several hormones related not only to abiotic but also to biotic stress in order to ascertain its putative involvement in a more general stress response.

In the past years NAC transcription factors have been identified in a wide range of model and also crop plants. Furthermore, their relevance in the integration of abiotic and also biotic stresses has also been addressed. However, most of the functional studies regarding the molecular mechanisms that could promote stress tolerance in the last instance have been performed preferentially in model plants such as A. thaliana. For this reason, I would like to emphasize that Yang and coworkers have made a step-forward in the evaluation of putative factors involved in drought stress tolerance with biotechnological applications, as the stress responses mediated by GmNAC12 have been analyzed entirely in soybean. As the authors mention in the manuscript (line 90 and line 289; reference paper number 38 (line 564)), a very similar study from the same laboratory  at Nanjing Agricultural University regarding the dissection of the stress response mediated by GmNAC8 (a GmNAC12 homolog)  in soybean was recently published (2020). Undoubtedly, the authors have produced very valuable tools to unravel the molecular mechanisms involved in the tolerance to stress of soybean plants.

The results shown by the authors regarding  the analysis of GmNAC12 expression under stress conditions seem to be solid, as it is the effect in drought tolerance observed in overexpression and knockout lines. Also the nuclear localization of this transcription factor is coherent with its putative function.

However, what surprises me most is that the authors do not go deeper in exploiting such biological tools to unravel the molecular mechanisms involved in stress tolerance. There are lots of open questions that could be assessed to improve the manuscript, such as:

- Molecular signals that could be involved in GmNAC12 expression (e.g., analysis of the promoter to search for putative TF binding motifs; evaluation of the expression of stress-related TFs (e.g., DREB, ARF, COR, etc.) under stress conditions and in mutated soybean lines).

- Search for putative targets of GmNAC12 action (e.g., in silico screening of NAC-specific recognition sequences in soybean genes and evaluate their expression patterns in correlation with GmNAC12 accumulation; ChIP assay; yeast one-hybrid assay, etc.).

- Physiological changes observed in the overexpressing lines that could contribute to their increased tolerance to drought (Stomatal closure? Lateral root formation? (several studies on NAC TFs point to this)).

- As drought and salt stresses share common signaling pathways…, are overexpressing lines also tolerant to salt stress?.

But what seems most important to me is to explore in more depth the results obtained in the yeast two-hybrid assay:

-        There is a considerable enrichment in interactor proteins related to biotic stress (bacterium, virus, fungus, etc.). Although the authors have demonstrated that GmNAC12 respond to hormones related to biotic stress, such as SA or MeJA, it does not clarify if GmNAC12 overexpression contributes to an increase or reduction in tolerance to pathogens. It would be interesting to address this issue with real infections in overexpression lines, in case an increase in salt tolerance could be associated with an increase in sensitivity to pathogens.

-        But my greatest concern relies on the fact that being GmNAC12 a transcription factor of nuclear localization its interaction partners should also be located in the nucleus. It is surprising to see such a huge number of interactors, most of them cytoplasmatic. Taking this into consideration, one would expect either:

o   Most of the interactions observed in yeast two-hybrid are artifacts.

o   GmNAC12 is also located in the cytoplasm (some previous studies have reported NAC TFs linked to membranes…, could it be possible for GmNAC12 to change its location under stress conditions?).

In my opinion, the authors should focus on interactors known to be TFs or nuclear-resident proteins for deeper studies of their putative roles in the response to drought stress.

The most important question is that, both in the abstract and in the discussion of the results, the authors claim that the molecular mechanism of GmNAC12 to improve drought tolerance rely on the antioxidant defense pathway (specifically on the activity of intracellular peroxidases). It can be true, but there is no analysis in the manuscript supporting this asseveration. Additional analysis should be performed in order to clarify this (e.g. co-localization assays, analysis of peroxidase activity or protein accumulation in mutated soybean lines, analysis of expression of related genes, etc.)

And the last, but not the least, minor mistakes observed in the manuscript should be revised before publication:

-        The citations to bibliography are completely disorganized (the number assigned in the text do not correspond to the paper in the bibliography list). Indeed, as an example, the authors call to their previous article concerning GmNAC8 characterization as number 39 (line 90, 289,324 and 331), while its position in the bibliographic section is 38. See also the reference in line 348.

-        Please, give a precise reference genome ID for GmNAC12 (e.g. Glyma.xxGxxxxxx)

Most of the opinions outlined in this report are suggestions that could be taken into consideration to improve the quality of the manuscript. However, those highlighted in bold should be modified before publication.

Author Response

Dear reviewer

We appreciate the critical comments and useful suggestions from you. All the questions and comments have been acknowledged. The manuscript has been also rewritten and organized as suggestions. Herein we re-submit our manuscript entitled “A Stress-related NAC Transcription Factor GmNAC12 from Soybean Positively Regulates Drought Stress Tolerance” (Manuscript ID: ijms-1935753) for consideration as article by International Journal of Molecular Sciences.

 Sincerely,

Prof. Jingming Zhao

Reviewer: Comments on the manuscript " A Stress-related NAC Transcription Factor GmNAC12 from Soybean Positively Regulates Drought Stress Tolerance ", by Yang et al.

  1. But my greatest concern relies on the fact that being GmNAC12 a transcription factor of nuclear localization its interaction partners should also be located in the nucleus. It is surprising to see such a huge number of interactors, most of them cytoplasmatic. Taking this into consideration, one would expect either:

o   Most of the interactions observed in yeast two-hybrid are artifacts.

o   GmNAC12 is also located in the cytoplasm (some previous studies have reported NAC TFs linked to membranes…, could it be possible for GmNAC12 to change its location under stress conditions?).

In my opinion, the authors should focus on interactors known to be TFs or nuclear-resident proteins for deeper studies of their putative roles in the response to drought stress.

Re: Using GmNAC12 as bait protein, 185 candidate interacting proteins were obtained by yeast library screening. For GmNAC12 transcription factors, via the website online (ProtComp - Predict the sub-cellular localization for Plant proteins (softberry.com)) to predict the protein expression in the plant cell nuclear, prediction results were validated by transient expression of tobacco leaf, but these results did not mean that under the any condition GmNAC12 were expressed in the nuclear in soybeans, for studies have shown that some transcription factors could change its expression under drought stress condition (Zhang S, et al. In rose, transcription factor PTM balances growth and drought survival via PIP2;1 aquaporin[J]. Nature Plants, 2019, 5(3):290-299). It had also been shown that the position of gene expression could be changed by protein interaction (Huang Y, et al. GmHsp90A2 is involved in soybean heat stress as a positive regulator[J]. Plant Science, 2019, 285:26-33), and the site prediction of protein expression position was not always accurate, it was possible to predict that the protein expressed in the cytoplasm would also be expressed in the nuclear. Therefore, proteins predicted to be in the cytoplasm were not excluded in the analysis of GmNAC12 interacting proteins in this study.

  1. The most important question is that, both in the abstract and in the discussion of the results, the authors claim that the molecular mechanism of GmNAC12 to improve drought tolerance rely on the antioxidant defense pathway (specifically on the activity of intracellular peroxidases). It can be true, but there is no analysis in the manuscript supporting this asseveration. Additional analysis should be performed in order to clarify this (e.g. co-localization assays, analysis of peroxidase activity or protein accumulation in mutated soybean lines, analysis of expression of related genes, etc.)

Re: The regulatory pathways of drought tolerance in plants are not only related to peroxide system but also related to other pathways. In the analysis of GmNAC12 interacting proteins in Result 2.5, we listed the pathways that might be related to drought stress tolerance without any emphasis. In the discussion, we used a paragraph to discuss the regulation of reactive oxygen species pathway on drought stress, because the previous research results of our laboratory found that GmNAC12 homolog GmNAC8 could improve the activity of peroxidase, but this did not mean that GmNAC12 has the same function in the positive regulation of soybean drought stress. We only wanted to list the experimental analysis results in this paper to provide reference for researchers, rather than focus on a particular regulatory pathway for further analysis. Therefore, we did not include relevant results in the paper.

  1. The citations to bibliography are completely disorganized (the number assigned in the text do not correspond to the paper in the bibliography list). Indeed, as an example, the authors call to their previous article concerning GmNAC8 characterization as number 39 (line 90, 289,324 and 331), while its position in the bibliographic section is 38. See also the reference in line 348.

After a brief introduction, you started talking about the research with results, while you should describe what you did at the first. We don’t know what your research is.

Re: Thank you. We have updated the references section.

  1. Please, give a precise reference genome ID for GmNAC12 (e.g. Glyma.xxGxxxxxx)

Re: We have put the reference genome ID for GmNAC12 in the first sentence of the last paragraph of the introduction.

“Previously, GmNAC12 (Glyma.16G043200) in soybean plants has been reported with induced up-regulated expression from stress caused by drought [25, 45].”

Round 2

Reviewer 2 Report

Reviewer: Comments on the manuscript " A Stress-related NAC Transcription Factor GmNAC12 from Soybean Positively Regulates Drought Stress Tolerance ", by Yang et al.

  1. But my greatest concern relies on the fact that being GmNAC12 a transcription factor of nuclear localization its interaction partners should also be located in the nucleus. It is surprising to see such a huge number of interactors, most of them cytoplasmatic. In my opinion, the authors should focus on interactors known to be TFs or nuclear-resident proteins for deeper studies of their putative roles in the response to drought stress.

Re: Using GmNAC12 as bait protein, 185 candidate interacting proteins were obtained by yeast library screening. For GmNAC12 transcription factors, via the website online (ProtComp - Predict the sub-cellular localization for Plant proteins (softberry.com)) to predict the protein expression in the plant cell nuclear, prediction results were validated by transient expression of tobacco leaf, but these results did not mean that under the any condition GmNAC12 were expressed in the nuclear in soybeans, for studies have shown that some transcription factors could change its expression under drought stress condition (Zhang S, et al. In rose, transcription factor PTM balances growth and drought survival via PIP2;1 aquaporin[J]. Nature Plants, 2019, 5(3):290-299). It had also been shown that the position of gene expression could be changed by protein interaction (Huang Y, et al. GmHsp90A2 is involved in soybean heat stress as a positive regulator[J]. Plant Science, 2019, 285:26-33), and the site prediction of protein expression position was not always accurate, it was possible to predict that the protein expressed in the cytoplasm would also be expressed in the nuclear. Therefore, proteins predicted to be in the cytoplasm were not excluded in the analysis of GmNAC12 interacting proteins in this study.

Re: Yes, It’s very interesting and it would be nice to include those explanations briefly in the text.

  1. The most important question is that, both in the abstract and in the discussion of the results, the authors claim that the molecular mechanism of GmNAC12 to improve drought tolerance rely on the antioxidant defense pathway (specifically on the activity of intracellular peroxidases). It can be true, but there is no analysis in the manuscript supporting this asseveration. Additional analysis should be performed in order to clarify this (e.g. co-localization assays, analysis of peroxidase activity or protein accumulation in mutated soybean lines, analysis of expression of related genes, etc.)

Re: The regulatory pathways of drought tolerance in plants are not only related to peroxide system but also related to other pathways. In the analysis of GmNAC12 interacting proteins in Result 2.5, we listed the pathways that might be related to drought stress tolerance without any emphasis. In the discussion, we used a paragraph to discuss the regulation of reactive oxygen species pathway on drought stress, because the previous research results of our laboratory found that GmNAC12 homolog GmNAC8 could improve the activity of peroxidase, but this did not mean that GmNAC12 has the same function in the positive regulation of soybean drought stress. We only wanted to list the experimental analysis results in this paper to provide reference for researchers, rather than focus on a particular regulatory pathway for further analysis. Therefore, we did not include relevant results in the paper.

Re: Yes, many different pathways may be involved in tolerance to drought stress. However, even though it is possible that the regulation of ROS by peroxidases may be important for drought tolerance, the authors do not include relevant results in this paper supporting that asseveration. Although those comments could be included in the results or the discussion, in my opinion they should be removed from the abstract (line 34) and the conclusions (line 458), as they are merely hypotheses and not proven results.

Author Response

Reviewer: Comments on the manuscript " A Stress-related NAC Transcription Factor GmNAC12 from Soybean Positively Regulates Drought Stress Tolerance ", by Yang et al.

1. Yes, It’s very interesting and it would be nice to include those explanations briefly in the text.

Re: Thank you for your advice. I have added a paragraph about this to the discussion.

We used GmNAC12 as bait protein and screened 185 candidate interacting proteins through Yeast two hybrid (Y2H). Interestingly, although GmNAC12 was a transcription factor expressed in the nuclear (Figure 4), its interacting proteins were expressed not only in the nuclear but also in the cytoplasm and cell membrane. It had been shown that the rose transcription factor PTM could change its expression position in response to drought stress [52], which might suggest that GmNAC12 was not exclusively expressed in soybean nuclear. In addition, GmHsp90A2 could change its protein expression region by interacting with GmHsp90A1 in soybean [9]. Therefore, GmNAC12 might change its expression position by interacting with proteins that function in different regions of the cell.

2. Yes, many different pathways may be involved in tolerance to drought stress. However, even though it is possible that the regulation of ROS by peroxidases may be important for drought tolerance, the authors do not include relevant results in this paper supporting that asseveration. Although those comments could be included in the results or the discussion, in my opinion they should be removed from the abstract (line 34) and the conclusions (line 458), as they are merely hypotheses and not proven results.

Re: Thank you for your advice. I have removed out the relevant content in the abstract and conclusion.

Abstract: NAC transcription factors (TFs) could regulate drought stresses in plants; however, the function of NAC TFs in soybeans remains unclear. To unravel NAC TFs function, we established that GmNAC12, a NAC TF from the soybean (Glycine max) was involved in the manipulation of stress tolerance. The expression of GmNAC12 was significantly up-regulated more than 10-fold under drought stress and more than 3-fold under abscisic acid (ABA) and ethylene (ETH) treatment. In order to determine the function of GmNAC12 under drought stress conditions, we generated GmNAC12 overexpression and knockout lines. The present findings showed that under drought stress, the survival rate of GmNAC12 overexpression lines increased by more than 57 % compared with wild-type plants, while the survival rate of GmNAC12 knockout lines decreased by at least 46 %. Furthermore, a subcellular localization analysis showed that the GmNAC12 protein is concentrated in the nucleus of the tobacco cell. In addition, we used a yeast two-hybrid assay to identify 185 proteins that interact with GmNAC12. Gene ontology (GO) and KEGG analysis showed that GmNAC12 interaction proteins are related to chitin, chlorophyll, ubiquitin-protein transferase, and peroxidase activity. Hence, we have inferred that GmNAC12, as a key gene, could positively regulates soybean tolerance to drought stress.

Conclusions: This study explored the potential functions of GmNAC12, a member of the soybean NAC family, in response to abiotic. The present results revealed that GmNAC12 overexpressed lines demonstrated improved tolerance to drought compared to wild-type plants, whereas GmNAC12 knockout lines were sensitive under drought conditions. Furthermore, the findings demonstrated that GmNAC12 positively regulates drought stress in soybeans. In addition, the 185 candidate interaction proteins of GmNAC12 identified by yeast two-hybrid assay were performed on GO analysis and KEGG enrichment, showing that some interaction proteins of GmNAC12 are related to peroxidase activity. Hence, we inferred that GmNAC12, as a key gene, could positively regulates soybean tolerance to drought stress.